# Development of a Mortality Prediction Model in Hospitalised SARS-CoV-2 Positive Patients Based on Routine Kidney Biomarkers

**DOI:** 10.3390/ijms23137260

**Published:** 2022-06-30

**Authors:** Anna N. Boss, Abhirup Banerjee, Michail Mamalakis, Surajit Ray, Andrew J. Swift, Craig Wilkie, Joseph W. Fanstone, Bart Vorselaars, Joby Cole, Simonne Weeks, Louise S. Mackenzie

**Affiliations:** 1School of Applied Sciences, University of Brighton, Brighton BN2 4GJ, UK; anna@boss-informatik.ch (A.N.B.); s.weeks@brighton.ac.uk (S.W.); 2Division of Cardiovascular Medicine, Radcliffe Department of Medicine, University of Oxford, Oxford OX3 9DU, UK; abhirup.banerjee@cardiov.ox.ac.uk; 3Department of Infection, Immunity and Cardiovascular Disease, University of Sheffield, Sheffield S10 2RX, UK; mmamalakis1@sheffield.ac.uk (M.M.); a.j.swift@sheffield.ac.uk (A.J.S.); joby.cole1@nhs.net (J.C.); 4School of Mathematics and Statistics, University of Glasgow, Glasgow G12 8QQ, UK; surajit.ray@glasgow.ac.uk (S.R.); craig.wilkie@glasgow.ac.uk (C.W.); 5Brighton and Sussex Medical School, Falmer Campus, University of Brighton, Brighton BN1 9PX, UK; j.fanstone1@uni.bsms.ac.uk; 6School of Mathematics and Physics, University of Lincoln, Brayford Pool, Lincoln LN6 7TS, UK; bvorselaars@lincoln.ac.uk

**Keywords:** SARS-CoV-2, prediction model, kidney function, COVID-19

## Abstract

Acute kidney injury (AKI) is a prevalent complication in severe acute respiratory syndrome coronavirus 2 (SARS-CoV-2) positive inpatients, which is linked to an increased mortality rate compared to patients without AKI. Here we analysed the difference in kidney blood biomarkers in SARS-CoV-2 positive patients with non-fatal or fatal outcome, in order to develop a mortality prediction model for hospitalised SARS-CoV-2 positive patients. A retrospective cohort study including data from suspected SARS-CoV-2 positive patients admitted to a large National Health Service (NHS) Foundation Trust hospital in the Yorkshire and Humber regions, United Kingdom, between 1 March 2020 and 30 August 2020. Hospitalised adult patients (aged ≥ 18 years) with at least one confirmed positive RT-PCR test for SARS-CoV-2 and blood tests of kidney biomarkers within 36 h of the RT-PCR test were included. The main outcome measure was 90-day in-hospital mortality in SARS-CoV-2 infected patients. The logistic regression and random forest (RF) models incorporated six predictors including three routine kidney function tests (sodium, urea; creatinine only in RF), along with age, sex, and ethnicity. The mortality prediction performance of the logistic regression model achieved an area under receiver operating characteristic (AUROC) curve of 0.772 in the test dataset (95% CI: 0.694–0.823), while the RF model attained the AUROC of 0.820 in the same test cohort (95% CI: 0.740–0.870). The resulting validated prediction model is the first to focus on kidney biomarkers specifically on in-hospital mortality over a 90-day period.

## 1. Introduction

Evidence suggests that the kidneys are involved in severe acute respiratory syndrome coronavirus 2 (SARS-CoV-2) infections directly and indirectly; acute kidney injury (AKI) is a prevalent complication in SARS-CoV-2 positive inpatients, which is linked to an increased mortality rate compared to patients without AKI [1]. It has been established that pulmonary system and gastrointestinal tract are primary entry sites of SARS-CoV-2 [2]. The virus is known to infect via binding to the transmembrane receptor angiotensin-converting enzyme 2 (ACE2), although this entry site may be less important for the SARS-CoV-2 Omicron variant (B.1.1.529) [3].

In the airways, studies of RNA in situ hybridisation in post-mortem kidneys found viral spike protein RNA in cells of kidney tubules [4]. ACE2 is highly expressed in podocytes and proximal straight tubule cells in the kidney [5], which indicates that the renal system is a possible site of infection, although most of the renal impairment is due to the impact of dehydration and hypovolaemia. Multi-organ involvement in SARS-CoV-2 infections is linked to systemic hyper-inflammation [6], and a large number of pro-inflammatory cytokines such as IL-6 induces tubular damage [7]. The resulting cell apoptosis affects kidney function and leads to an increase in leukocytes infiltration [4].

AKI caused by hypovolaemia, viral infection, and nephrotoxic drugs [8] is a common complication in hospitalised patients affecting 10% to 20% of inpatients [9]. Many studies reported varying rates of AKI in SARS-CoV-2 infection ranging from 5.1% [10] to 36.6% [11]. Furthermore, SARS-CoV-2 patients who develop AKI appear to have an increased risk of mortality compared to patients without kidney involvement [12]. The estimated mortality rate in all SARS-CoV-2 patients is about 2% to 3% [13], and the mortality rates in SARS-CoV-2 positive patients who developed AKI were reported to be as high as 80% [1]. In some small-scale analysis of patient data, increased creatinine and urea levels are associated with mortality [14]. Moreover, SARS-CoV-2 positive patients with fatal outcome experience decreased levels of sodium, potassium, and bicarbonate [15]. While this may be an indirect correlation, it is apparent that kidney function has an important role in severity and outcome for the patient.

The current pandemic urgently needs support in medical decision making and triaging to allocate the limited resources to enable healthcare workers to assess patient status and determine whether the patient should be better treated outside of the hospital environment [16]. The majority of the prediction models developed for Coronavirus disease 2019 (COVID-19) have an easy-to-use score; there is often a high risk of bias, which impedes the clinical implementation. Thus, transparent reporting is important to facilitate external validation of developed models.

Numerous prediction models for COVID-19 mortality have been developed [17,18,19,20], yet there has been no model to our knowledge, exclusively based on kidney blood biomarkers and patient characteristics. Previous studies suggest that the kidneys are involved in SARS-CoV-2 infections, and AKI patients have an increased risk of mortality. Given the ongoing COVID-19 pandemic and globally rising number of deaths, it is important to understand the kidney damage caused by SARS-CoV-2 infection. This will still be important when SARS-CoV-2 is endemic. Additionally, understanding pathophysiological mechanisms of the kidney in SARS-CoV-2 infection may guide pharmacotherapy in SARS-CoV-2 positive patients to avoid drugs that are potentially harmful to the kidneys, such as non-steroidal anti-inflammatory drugs (NSAIDs), antibiotics, or antiviral drugs.

In this work, we explored the involvement of the kidneys in SARS-CoV-2 infections with routine blood biomarkers of renal function in a retrospective cohort study. We aimed to establish a mortality prediction model based on routine kidney function tests measured in early SARS-CoV-2 infection.

## 2. Results

### 2.1. Participants

The data extracted from Laboratory Information Management System (LIMS) comprised 2314 SARS-CoV-2 real-time polymerase chain reaction (RT-PCR) test results collected between 1 March 2020 and 30 August 2020 from 1165 SARS-CoV-2 positive or negative patients (Figure 1). The number of SARS-CoV-2 positive patients after excluding all multiple SARS-CoV-2 RT-PCR tests and patients who did not fulfil the inclusion criteria was 788 (289 fatal and 499 non-fatal cases).

The timeframe of mortality was determined by the mean time span ± 2 standard deviation (SD) between the first positive SARS-CoV-2 RT-PCR test up to the day of death (21.9 days ± 34.4 days; Figure 2). Deaths occurring after 90 days were excluded from the subsequent analysis. 

### 2.2. Non-Fatal and Fatal Outcomes in SARS-CoV-2 Positive Patients

Table 1 shows the difference in SARS-CoV-2 positive patients with and without fatal outcomes for all candidate predictors. Among SARS-CoV-2 positive patients, the majority were male, both in the non-fatal (54.7%) and fatal (63.0%) groups. Moreover, fatal outcome was significantly more prevalent in male patients than in female patients. Median age of the fatal group was 81 years (Interquatile range; IQR: 75–87), whereas median age for the non-fatal group amounted to 69 years (IQR: 55–80) and was thus significantly lower (*p* < 0.001).

There was a significant increase in sodium (*p* < 0.001) and urea (*p* < 0.001) blood levels in the fatal group compared to the non-fatal group. The increase in creatinine levels was significantly high in the fatal group as well as in the male patients, but not for the female SARS-CoV-2 positive patients with fatal outcome.

### 2.3. Prediction Models

The steps and methods for dataset pre-processing, primary model fitting to develop the models, and the final evaluation are outlined in a flow diagram (Figure 3).

### 2.4. Missing Values

After testing the level of completeness of the 10 candidate predictors, chloride (0.1%) and bicarbonate (7.3%) were excluded from the model due to their high level of missingness (Appendix A). For all other candidate predictors, at least 72.7% of patients had a recorded measurement. We used multiple imputation by chained equations (MICE) to replace the missing values.

### 2.5. Candidate Predictors Selection

We selected the predictors sex, age, sodium, and urea for inclusion in the prediction model due to the significant difference between non-fatal and fatal outcomes (Table 1). Even though not all subgroups of ethnicity differed significantly between non-fatal and fatal outcomes, the overall *p*-value for all ethnicities together was below 0.05. Therefore, we decided to include ethnicity as a predictor in the models. Creatinine was significantly different between the non-fatal and fatal groups for male patients and both sexes combined, but not for females. Despite being dependent on sex, we decided to include creatinine as a candidate predictor, because over 50% of female patients in both groups had increased creatinine levels indicating its importance in a SARS-CoV-2 infection. Potassium was not significantly different in non-fatal and fatal outcome groups and was thus excluded from the model. Estimated glomerular filtration rate (eGFR) could not be included in the model because of its multi-collinearity with other predictors.

### 2.6. Model Development

#### 2.6.1. Logistic Regression

The logistic regression model included the selected six predictors, namely sex, age, ethnicity, sodium, urea, and creatinine. After applying the cross-validated L1 regularisation of 0.70 to the model, creatinine was excluded from the model. The L1 regularisation predictor selection clarified the dependency of creatinine on sex, thus the low importance in predicting mortality in SARS-CoV-2 positive patients. Age had the greatest weight (5.01) in the model (Table 2). Urea and sodium followed with 2.83 and 1.43, respectively. The predicted probability of 90-day in-hospital mortality can be measured by the following expression:(1)$$P_{\mathrm{Mortality}}=\frac{1}{1+e^{-\left(-5.31+0.53*\mathrm{Sex}+5.01*\frac{\left(\mathrm{Age}-20\right)}{82}-0.40*\mathrm{Ethnicity}+1.43*\frac{\left(\mathrm{Sodium}-107\right)}{60}+2.83*\frac{\left(\mathrm{Urea}-1.3\right)}{49.9}\right)}}$$

In the prediction model, values of 0 and 1 were used for Female and Male, respectively. For ethnicity, the following values were assigned: 0.0 for White, 0.2 for Mixed, 0.4 for Asian, 0.6 for Black, 0.8 for Chinese, and 1.0 for any other ethnicity.

#### 2.6.2. Random Forest

Stratified 5-fold cross-validation determined that 41 trees were optimal, a tree depth of 8 was ideal, and optimal minimum number of samples required to split an internal node was 0.1. Urea was the most important predictor followed by age (Appendix A). Ethnicity and sex were the two least important predictors in the model with 10% and 3% of the importance compared to urea, respectively.

#### 2.6.3. Simplified Model

The third model was a simplified version of the first logistic regression model and only included the predictors sex, age, and ethnicity. We applied a regularisation strength of 0.87. The coefficients of sex, age, and ethnicity were 0.59, 5.42, and −0.37, respectively with an intercept of −4.43.

### 2.7. Model Performance in Training Cohort

After developing the three models, we evaluated their discriminative power in the training cohort. The logistic regression model showed good discrimination for in-hospital mortality with an AUROC of 0.769 (95% CI: 0.716–0.821), as seen in Table 3. The random forest model had a greater AUROC of 0.800 (95% CI: 0.750–0.820), whereas the simplified model (only patient characteristics as predictors) had an AUROC of 0.748 (95% CI: 0.700–0.794).

Sensitivity was calculated as $\frac{\mathrm{True}\;\mathrm{positive}}{\left(\mathrm{True}\;\mathrm{positive}+\mathrm{False}\;\mathrm{negative}\right)}$. Specificity was calculated as $\frac{\mathrm{True}\;\mathrm{negative}}{\left(\mathrm{True}\;\mathrm{negative}+\mathrm{False}\;\mathrm{positive}\right)}$. AUROC: area under receiver operating characteristic curve.

The slope of the calibration curve for the logistic regression model was excellent with 0.986. The random forest classifier had a calibration slope of 1.093 and the simplified model had a slope of 1.127. The Brier score was comparable in all three models (Table 3). The specificity value for the logistic regression was high with 0.838. The random forest and simplified model also had similar specificity of 0.825. In contrast, the sensitivity was lower for all three models.

### 2.8. Model Evaluation in Test Cohort

The power of performance of the logistic regression model in the test cohort (Table 3) showed an AUROC of 0.772 (95% CI: 0.694–0.823; Appendix A). The random forest model achieved the AUROC value of 0.820 (95% CI: 0.740–0.870; Appendix A) in the same cohort, while the simplified model attained the AUROC of 0.757 (95% CI: 0.678–0.815; Appendix A). Calibration measures were similar in the logistic regression and random forest model with 1.109 for the logistic regression and 1.190 for the random forest (Appendix A and Table 3). The simplified model had a slope of 0.846, which means the model tended to overestimate the probability of mortality. Brier Score and specificity were good and comparable in all three models. Sensitivity was still low in all models, ranging from 0.414 to 0.483.

The four risk groups were defined with the cut-off values −2.96, −0.62, and 1.73 (Appendix A). The corresponding mortality rates were <5% for low-risk, 5% to 35% for intermediate-risk, 35% to 85% for high-risk, and >85% for very high-risk. Most patients (54.8%) were classified at a high-risk of a fatal outcome (Table 4). Very few (0.8%) patients were classified into the group of very high-risk.

## 3. Discussion

We compared patient characteristics and kidney blood biomarkers of SARS-CoV-2 positive patients with non-fatal outcome to those of patients with fatal outcome. We found that male sex, increasing age, and the ethnicities White and Black were significantly associated with fatal outcome in SARS-CoV-2 infection. Few studies found no significant link between sex and fatal outcome [21,22]. In contrast, An et al. [23] in their study over 10,237 patients found that male patients are significantly more likely to have a fatal outcome than female patients from a SARS-CoV-2 infection. There are many different hypotheses for why males could be more prone to a fatal outcome [24]. For example, gender differences in immune responses and susceptibility to viral infections have been seen. Moreover, behavioural and social differences between female and male patients, such as smoking and alcohol consumption, could contribute to a significant difference in fatal outcome.

We have observed increased sodium, urea, and creatinine levels (in males), as well as reduced eGFR, are linked to a fatal outcome in a SARS-CoV-2 infection, in keeping with other studies [21,22]. However, Bonetti et al. [21] did not find any significant differences in electrolytes between SARS-CoV-2 patients with fatal outcome and those with non-fatal outcome. This difference might be explained by the lower patient number (n = 144) in their study.

Our findings suggest that kidney involvement in SARS-CoV-2 positive patients differs between the groups with non-fatal and fatal outcomes. Although the patients with fatal outcome had normal electrolyte levels, the levels of urea, creatinine, and eGFR were outside the reference ranges indicating kidney impairment.

We developed three prediction models using LASSO logistic regression and random forest for in-hospital mortality in a retrospective cohort study with 788 SARS-CoV-2 positive patients. The models included three to six predictors routinely available at hospital admission, such as patient characteristics and routine kidney function tests. Age was an important predictor of mortality in all three models. In contrast, the relative weights of the predictors ethnicity and sex were small in all models. Interestingly, creatinine was excluded from the logistic regression model, but was the third most important predictor in the random forest model. Comparing our models to other COVID-19 mortality prediction models, age seemed to be consistently one of the most important predictors [19,23].

Performance of both our logistic regression and random forest model was good. Even though logistic regression had a lower discrimination power than random forest, the greater interpretability might be important. Interpretability and simple application of prediction models are key factors in clinical use. Loss in discrimination is justified and acceptable to enhance the use of the model in clinical settings.

As age had the largest weight in the logistic regression model, it was no surprise that the simplified model performed well with an area under receiver operating characteristic (AUROC) curve of 0.757. Including the predictors urea and sodium improved the discriminative power of the model. This suggests that kidney biomarkers that indicate the state of kidney health add valuable discrimination power to the mortality prediction.

To the extent of the authors’ knowledge, this study is the first to focus solely on routine kidney function tests in combination with patient characteristics and the prediction of fatal outcome from SARS-CoV-2 infections. We did not find any existing model with the same combination of predictors. Knight et al. [19] developed the 4C Mortality Score with aims similar to those in our models. Their LASSO model was built with data of 35,463 patients using age, sex, comorbidities, respiratory rate, peripheral oxygen saturation, Glasgow coma scale, urea, and C-reactive protein (CRP) as predictors. Therefore, our model shares three predictors with the 4C Mortality score which had an AUROC of 0.767 (95% CI: 0.760–0.773) in the test cohort. This is similar to our logistic regression model (0.772); however, our model requires only five predictors which are arguably simpler to obtain.

A previous study from our group developed a logistic regression model named LUCAS [18], based on 1434 patients and using five routinely available predictors: Lymphocyte, Urea, CRP, Age, and Sex. The discrimination power of the LUCAS model and our logistic regression model are similar (0.765 vs. 0.772). However, the sensitivity of the LUCAS model was markedly higher (0.931) than that of our model (0.414). In contrast, our model has a markedly better specificity (0.820 vs. 0.330). Although the lower sensitivity of our model indicates that some SARS-CoV-2 positive patients are not accurately identified to have fatal outcome, the specificity of the model shows great performance, indicating 82 out of 100 SARS-CoV-2 positive patients with non-fatal outcome are correctly identified based only on their kidney biomarkers and patient characteristics.

Fundamental limitations of this study were the selection criteria of patients and the limited sample size of 788 patients. The selection of patients was based on the criteria defined by Ray et al. [18]. Only patients who had a chest x-ray examination were extracted from LIMS, which introduced bias in patient selection. We included just the required number of patients to limit overfitting of the model. However, certain subgroups had very small sample sizes. Additionally, only few patients were classified into the low-risk or very high-risk groups. This reduces the power of the calculated mortality rates in these groups.

A retrospective cohort study design carries inherent limitations such as missing values and unclear recording conditions. We carefully considered and minimised the risks of bias. For example, MICE was used to compensate for missing values. Comorbidities such as hypertension, chronic kidney disease, and diabetes can increase the risk of AKI, thus increasing the risk of abnormal levels of kidney function biomarkers [8]. Additionally, intake of NSAIDs or blood pressure medication increases the risk to develop AKI [8]. We could not use comorbidities or medication history as selection criteria since this information was missing in the extracted data.

We assumed missing information in certain cases, which limits the accuracy of the model. For example, the dataset did not indicate the causes of death. Therefore, we assumed that patients died due to the SARS-CoV-2 infection. We tried to address this assumption by including only patients who died within a defined timeframe. Most of the mentioned limitations can be avoided or reduced by using a prospective study design.

Even though we mentioned missing medical and medication history as a limitation of this study, we also consider it as a strength. Obtaining a complete medical history is time-consuming and might be difficult during a pandemic with limited resources. Our model, however, does not rely on this patient information to accurately predict in-hospital mortality. Moreover, we exclusively used routinely measured and readily available predictors at hospital admission, thus increasing the model’s clinical applicability. Another strength of this study is the adherence to the transparent reporting of a multivariable prediction model for individual prediction or diagnosis (TRIPOD) guidelines for the development and reporting of the prediction models.

The evaluation of our model on the test cohort confirmed the robustness of our model. External validation would further increase the robustness and generalisability. A scoring system could enhance applicability and clinical usefulness of the model. The model was developed based on data obtained during the first wave of the pandemic. The situation is changing rapidly, and treatment for SARS-CoV-2 infection, vaccination, and virus mutation are likely to alter the demographics of patients with fatal outcome. Therefore, updating and adapting the algorithm of the model is important to maintain accurate mortality prediction.

## 4. Materials and Methods

### 4.1. Study Design and Setting

We conducted a retrospective cohort study using data extracted from the LIMS. The patient data were collected at a large National Health Service (NHS) Foundation Trust hospital in the Yorkshire and Humber regions, United Kingdom.

### 4.2. Data Protection/Ethics

De-identified and pseudo-anonymised patient data were obtained from datasets, and the methods used were approved by the ethics committee as part of the existing Cardiac Magnetic Resonance Imaging (MRI) Database NHS Research Ethics Committee (REC) Integrated Research Application System (IRAS) Ref: 222,349 and University of Brighton REC (8011). The need for informed consent was waived by the ethics committee (University of Brighton REC) due to retrospective nature of the study. We followed the TRIPOD guidelines for model development and reporting [25].

### 4.3. Participants

The extracted dataset included 2314 SARS-CoV-2 RT-PCR tests. Patients with suspected SARS-CoV-2 infection were consecutively admitted to hospital between 1 March 2020 and 30 August 2020.

### 4.4. Inclusion and Exclusion Criteria

We used data of patients aged 20 to 102 years who had at least one confirmed positive RT-PCR test for SARS-CoV-2. We included patients with a blood test for kidney biomarkers within a 36-h window from the SARS-CoV-2 RT-PCR test to ensure that the results were representative of an early SARS-CoV-2 infection. If multiple positive RT-PCR results accompanied by laboratory tests were recorded for the same patient, only the earliest recording was used for model development. Patients without mortality information in the dataset were excluded from the analysis.

### 4.5. Mortality Timeframe

We defined the timeframe of mortality prediction using the curve of normal distribution. To retain 95% of all fatal outcome cases, we registered each patient who died within the time of two SD from the mean rounded to the closest multiple of five measured. Patients who died outside this period were excluded from the model.

### 4.6. Candidate Predictors

As the prediction model focuses on routinely available kidney function tests, candidate predictors included sodium (mmol/L), potassium (mmol/L), chloride (mmol/L), bicarbonate (mmol/L), urea (mmol/L), creatinine (μmol/L), and eGFR (mL/min/1.73 m^2^), along with age (years), sex, and ethnicity. Ethnicity subgroups were combined into six categories, namely White, Black, Asian, Mixed, Chinese, and any other ethnicity. Other important markers in SARS-CoV-2 positive patients are predictors for pneumonia such as oxygen saturation and C-reactive protein [26]. However, these predictors were not included in this study due to its focus on the kidney function.

### 4.7. Non-Fatal and Fatal Outcomes in SARS-CoV-2 Positive Patients

The medians of continuous predictors in SARS-CoV-2 positive cases with fatal outcome were compared with those in positive cases with non-fatal outcome using non-parametric Mann-Whitney U-test. Results were presented as median ± interquartile range (IQR). The categorical predictors sex and ethnicity were compared with a chi-squared test and reported as an absolute number of patients and percentages.

### 4.8. Prediction Models

#### 4.8.1. Outcome

The primary outcome of interest was 90-day in-hospital mortality in SARS-CoV-2 positive patients based on routine kidney function tests and patient characteristics. We were interested in the performance of a model in predicting mortality at the time of the first positive SARS-CoV-2 RT-PCR test.

#### 4.8.2. Model Development

We excluded patients with missing data for the SARS-CoV-2 RT-PCR test or mortality from the analysis to ensure data integrity. Only candidate predictors with a completeness level of 60% or more were kept for model development. Missing values were considered missing at random and handled by the multiple imputation by chained equations (MICE) [27]. We started with the predictor with the fewest missing values being set as output. After 10 imputation rounds, the resulting dataset was returned and rounded to two decimal places. We included all candidate predictors that were significantly different (*p* < 0.05) between fatal and non-fatal cases in the model development.

The imputed data were randomly split into stratified training and test sets at a ratio of 4 to 1 for all developed models. Before the models were trained, we scaled the data with min-max normalisation. We used least absolute shrinkage and selection operator (LASSO) multivariable logistic regression to develop the model. The regularisation strength λ of the penalty L1 was determined by stratified 5-fold cross-validation using λ with the maximum AUROC for the model.

We developed a second model with the machine learning method random forest to compare its performance to the logistic regression model. We tuned the number of trees, maximal depth of trees, and minimum number of samples required to split an internal node by stratified 5-fold cross-validation, selecting the model with the highest AUROC score.

We developed another simplified LASSO logistic regression model that only included patient characteristics (sex, age, and ethnicity) as candidate predictors. Regularisation strength λ of the penalty L1 was determined as previously described.

#### 4.8.3. Model Evaluation in Training Cohort

After developing the three prediction models, we examined the power of performance of each model in the training cohort with the measures of calibration and discrimination by stratified 5-fold cross-validation. Calibration was assessed with a calibration plot and the slope of the calibration curve in the training cohort. Discrimination was evaluated with the AUROC with 95% confidence intervals. Moreover, we calculated the Brier score, sensitivity, and specificity to assess the performance of the developed models.

#### 4.8.4. Model Evaluation in Test Cohort

The model was evaluated in the test cohort with the same measures used for the training cohort. The final discrimination and calibration values were presented as mean.

We defined four risk groups (low, moderate, high, and very high) with the logistic regression model. These groups can help clinical practitioners to quickly classify patients into different mortality risk groups. Grouping was created by dividing the logistic regression curve into four equal segments. We defined the mortality rates by calculating the percentage of patients with fatal outcome for each risk group.

We employed the packages scikit-learn (version 0.24.1) created by Fabian Pedregosa et al., Rocquencourt, France, [27], pandas (version 1.2.3) created by Wes McKinney Greenwich, Connecticut, United States [28], and matplotlib (version 3.4.1) created by John D. Hunter, Salt Lake, Utah, USA [29] in Python 3.9.1 created by Guido van Rossum, Amsterdam, The Netherlands for data loading, data processing, data analysis, and data visualisation.

## 5. Conclusions

We assessed the differences in kidney function tests and patient characteristics between SARS-CoV-2 positive patients with non-fatal or fatal outcome. Among the analysed predictors, all but potassium, bicarbonate, and chloride showed a significant link to fatal outcome. Furthermore, we have observed that Male sex, increasing age, and both White and Black ethnicities were significantly associated with a fatal outcome in SARS-CoV-2 infection.

Age contributed significantly to the logistic regression model, and the combination of the urea and sodium predictors improved the discriminative power of the model. This suggests that kidney biomarkers that indicate the state of kidney health can contribute importantly to the mortality prediction of SARS-CoV-2 infection.

To the best of our knowledge, this is the first study that focusses on kidney biomarkers specifically for developing a successful mortality prediction model for patients at an early stage of SARS-CoV-2 infection. Furthermore, we successfully developed a logistic regression mortality prediction model for in-hospital SARS-CoV-2 positive patients, based on levels of urea and sodium, as well as on age, sex, and ethnicity.

## Figures and Tables

**Figure 1 ijms-23-07260-f001:**
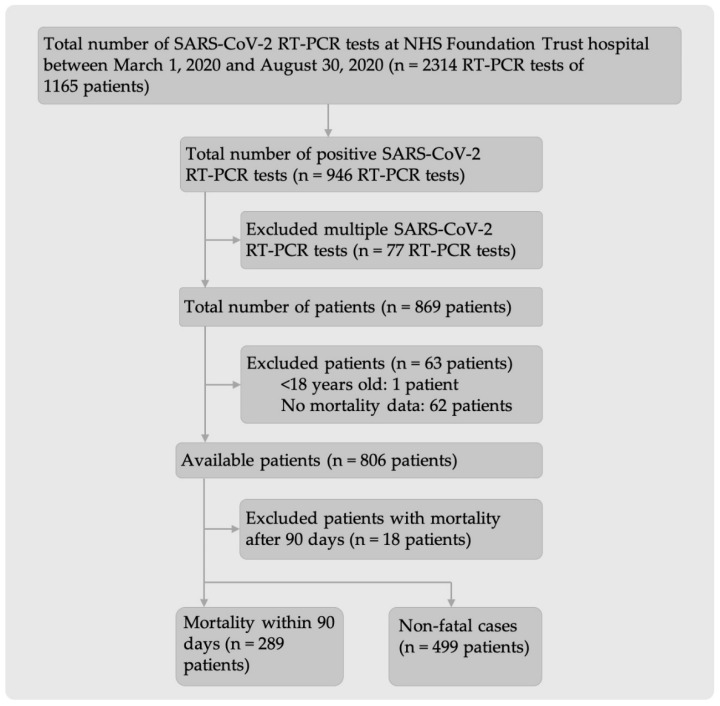
Flow diagram of inclusion and exclusion criteria for patients admitted to hospital with either suspected SARS-CoV-2 infection or before the COVID-19 pandemic.

**Figure 2 ijms-23-07260-f002:**
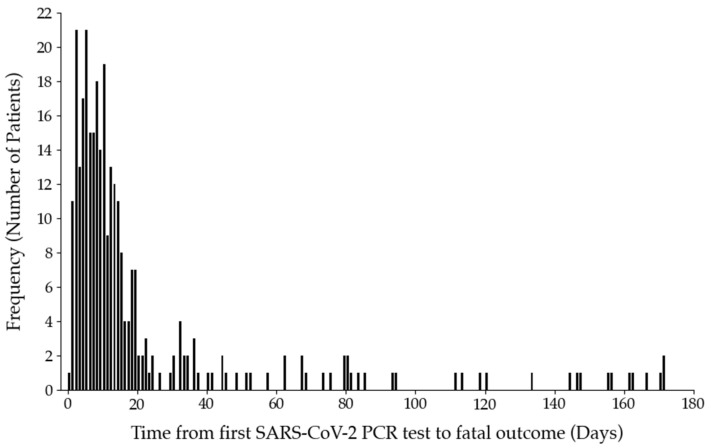
Histogram illustrating the time span from the first positive SARS-CoV-2 RT-PCR test of patients admitted to hospital until the day of death (n = 307). All patients within two standard deviations (mean = 21.9 days; SD = 34.4 days) were included in the study (n = 289). The remaining fatal cases after 90 days were excluded (n = 18).

**Figure 3 ijms-23-07260-f003:**
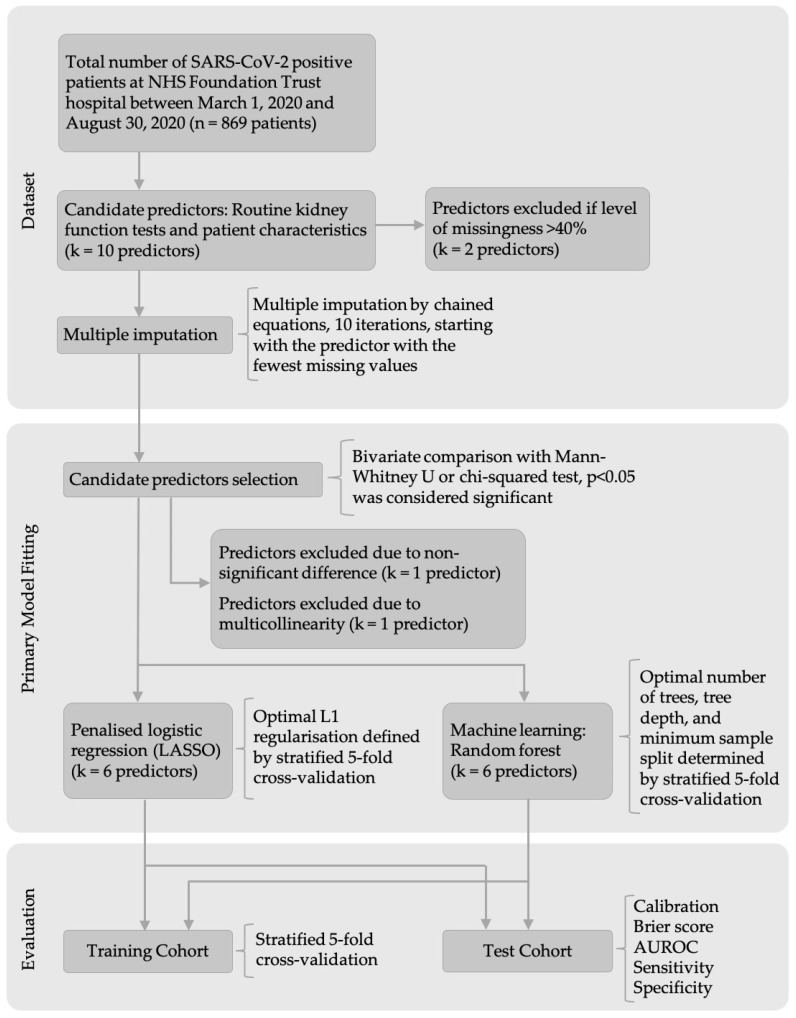
Model development and evaluation workflow.

**Table 1 ijms-23-07260-t001:** Demographics and kidney function characteristics.

Characteristics	Reference Range	Non-Fatal (n = 499)	Fatal (n = 289)	*p*-Value
		Median (IQR) or No. (%)	Median (IQR) or No. (%)	
Sex (%):	-			0.03
Male		273 (54.7)	182 (63.0)	
Female		226 (45.3)	107 (37.0)	
Age [years]	-	69.0 (55.0–80.0)	81.0 (75.0–87.0)	<0.001
Ethnicity (%):	-			<0.001
White		377 (75.6)	254 (87.9)	<0.001
Mixed		4 (0.8)	0 (0.0)	0.31
Asian		27 (5.4)	7 (2.4)	0.07
Black		38 (7.6)	8 (2.8)	<0.05
Chinese		0 (0.0)	1 (0.3)	0.78
Any other ethnicity		53 (10.6)	19 (6.6)	0.10
Sodium [mmol/L]	133.0–146.0	136.0 (133.0–138.0)	138.0 (134.0–140.0)	<0.001
Potassium [mmol/L]	3.5–5.3	4.2 (3.9–4.5)	4.2 (3.8–4.6)	0.22
Bicarbonate [mmol/L]	22.0–29.0	21.5 (18.0–24.2)	19.0 (18.0–24.0)	0.23
Chloride [mmol/L]	95.0–108.0	96.0 (-)	-	-
Urea [mmol/L]	2.5–7.8	6.1 (4.3–9.3)	9.6 (6.9–14.0)	<0.001
Creatinine [μmol/L]		84.5 (71.0–114.0)	101.0 (76.3–152.5)	<0.001
Male	62.0–106.0	90.0 (78.0–121.0)	117.0 (89.5–170.5)	<0.001
Female	44.0–80.0	76.0 (62.0–99.0)	82.0 (62.5–125.0)	0.08
eGFR [mL/min/1.73 m^2^]	≥60.0	73.0 (50.0–90.0)	51.0 (33.0–75.0)	<0.001

Difference between fatal and non-fatal groups with continuous variable was determined with a Mann-Witney U-test. Bivariate comparison of categorical variables sex and ethnicity was conducted using the chi-squared test, where a *p*-value of <0.05 was considered significant. Chloride could not be compared since the fatal group had no recorded measurements for this biomarker. IQR: interquartile range; eGFR: estimated glomerular filtration rate.

**Table 2 ijms-23-07260-t002:** Logistic regression penalised coefficients and scaling factors.

Predictor	Penalised Coefficient	Scaling Factor
Intercept	−5.31	-
Sex	0.53	-
Age [years]	5.01	(Age − 20)/82
Ethnicity	−0.40	-
Sodium [mmol/L]	1.43	(Sodium − 107)/60
Urea [mmol/L]	2.83	(Urea − 1.3)/49.9
Creatinine [μmol/L]	0.00	(Creatinine − 15)/814

Regression coefficients were obtained with a L1 regularisation factor λ of 0.70.

**Table 3 ijms-23-07260-t003:** Model performance in training and test cohorts.

Model Performance	Training Cohort			Test Cohort		
	Logistic Regression	Random Forest	Simplified Model	Logistic Regression	Random Forest	Simplified Model
Calibration	0.986	1.093	1.127	1.109	1.190	0.846
Brier score	0.188	0.176	0.193	0.185	0.170	0.191
AUROC (95% CI)	0.769 (0.716–0.821)	0.800 (0.750–0.820)	0.748 (0.700–0.794)	0.772 (0.694–0.823)	0.820 (0.740–0.870)	0.757 (0.678–0.815)
Sensitivity	0.500	0.522	0.457	0.414	0.466	0.483
Specificity	0.838	0.825	0.825	0.820	0.900	0.830
No. of true positive	23	24	21	24	27	28
No. of true negative	67	66	66	82	90	83
No. of false positive	13	14	14	18	10	17
No. of false negative	23	22	25	34	31	30

**Table 4 ijms-23-07260-t004:** Risk groups.

Risk Group	Probability of Fatal Outcome	Number of Patients (%)	Number of Fatal Cases (%)
Low	<0.05	51 (6.5)	1 (0.1)
Intermediate	0.05–0.35	311 (39.5)	58 (7.4)
High	0.35–0.84	420 (53.3)	225 (28.6)
Very high	>0.85	6 (0.8)	5 (0.6)

Total number of patients included was 788 patients.

## Data Availability

The dataset used for model development is not publicly available, and its use requires a license. The final model is freely accessible as a web tool on https://mdscore.org [30].

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
