# Peer review of "Development of a Mortality Prediction Model in Hospitalised SARS-CoV-2 Positive Patients Based on Routine Kidney Biomarkers"

_ijms, 2022, doi:10.3390/ijms23137260_

Round 1

Reviewer 1 Report

In the manuscript: “Development of a mortality prediction model in hospitalised SARS-CoV-2 positive patients based on routine kidney biomarkers”, the authors investigated in a retrospective cohort study, the involvement of the kidneys in SARS-CoV-2 infections with routine blood biomarkers of renal function.

The article is interesting and had very promising results regarding the health crisis that affected the whole world for more than two years.

I would recommend to do the following remarks and accept the manuscript after minor changes:

  1. Try to reorganize the Abstract, as it is very important for the visibility of the article. E.g., some data can be removed (lines 28-29).
  2. Line 251 – the full word for AUROC should be included.
  3. Line 141 – the full word for eGFR should be also added here as is the first time the abbreviation apereas in the text.
  4. Be careful to the expression from line 168. You should re-write it and also give it a number, as is mentioned in template.
  5. The Conclusions section are rather small compared to the other paragraphs.
  6. Please pay attention to the text processing:

- justify must be used in the whole manuscript;

- some extra spaces must be deleted (e.g., line 169, line 219);

- line 386 – a number for this sub-section should be probably added.

Reviewer 2 Report

Dear editors:  

 It is a great honor and pleasure for me to be invited as a reviewer for this work. Anna Boss and co-authors develop a mortality prediction model in hospitalized SARS- CoV-2 positive patients based on routine kidney biomarkers. Although the topic is important, I have a number of comments concerning the publication of this study in International Journal of Molecular Sciences:

  1. From the perspective of machine learning, this model was initially fit on a training cohort. Successively, the fitted model was applied to predict the responses for the observations in a second data set called the validation data sheet. Finally, the test cohort should be used to provide a promising prediction of a final model fit on the training cohort.
  2. Overall, the routine kidney function tests used in the current model are lack of novelty, along with age, sex, and ethnicity. Line 136 “Creatinine was only significantly different between the non-fatal and fatal groups for male patients, but not for females.” Since creatinine is not an independent predictor, it should not be included in the death prediction model.
  3. In light of the disease nature of SARS- CoV-2, author should display all of the clinical predictors for pneumonia, such as SpO2, O2 demand, CRP, etc.
  4. International Journal of Molecular Sciences is a reputable scholarly journal about molecular medicine with a high impact factor (5.92). Although the paper is well-written, the investigation does not fall within the scope.
  5. International Journal of Environmental Research and Public Health and Healthcare are interdisciplinary journals that may be suitable for publication.
  6. Author should explain why male patients were prone to suffer from SARS-CoV-2
  7. Likewise, why were both fatal and non-fatal outcomes significantly more prevalent in male patients than in female patients?
  8. Some typo errors were repeatedly noted, eg. uniform the term “outcomes or outcome”; a redundant space in “strength  of the penalty L1”.

          Thank you for giving me the opportunity to review this article.

Sincerely,

Reviewer 3 Report

Reviewing the manuscript entitled, “Development of a mortality prediction model in hospitalized SARS- 2 CoV-2 positive patients based on routine kidney biomarkers” by Boss A er al., this is an article focusing on development of a mortality prediction model in hospitalized SARS-Cov-2 infected patients. Three models were developed and markers of renal dysfunction were proposed as predictors. Although this is an interested manuscript, the authors need to respond to the following concerns.

Although you mentioned “In contrast, the sensitivity was lower for all three models. at line 202 and table 6, you need to describe an interpretation of these results in the discussion. In general, sensitivity refers to the probability that a patient will be correctly positive. If the sensitivity is increased, the number of false positives will increase. As in the case of cancer screening, I think that it is better to treat the suspicious patient as a patient requiring attention in this case.

Although you described in the discussion, the result is different from the previous papers, and the number of n in your experiment is small. The authors need to describe the medical validation regarding the developed model and results in the discussion.

Although the measures differ in each country, SARS-Cov-2 may be no longer a pandemic, but an endemic. So, the author needs to describe how to utilize this model.

The authors need to add an abbreviation table.
